

# Applicability of the ReproQ client experiences questionnaire for quality improvement in maternity care

Marisja Scheerhagen[1,2], Henk F. van Stel[3], Dominique J.C. Tholhuijsen[4], Erwin Birnie[2], Arie Franx[5] and Gouke J. Bonsel[1,2]

[1] Department of Obstetrics and Gynecology, Erasmus Medical Centre, Rotterdam, The Netherlands
[2] Department of Obstetrics and Gynecology, Academic Collaborative Maternity Care, University Medical Center Utrecht, Utrecht, The Netherlands
[3] Julius Center for Health Sciences and Primary Care, Department of Health Technology Assessment, University Medical Center Utrecht, Utrecht, The Netherlands
[4] Institute of Health Policy and Management, Erasmus University Rotterdam, Rotterdam, The Netherlands
[5] Department of Obstetrics and Gynecology, University Medical Center Utrecht, Utrecht, The Netherlands

## ABSTRACT

**Background.** The ReproQuestionnaire (ReproQ) measures the client's experience with maternity care, following the WHO responsiveness model. In 2015, the ReproQ was appointed as national client experience questionnaire and will be added to the national list of indicators in maternity care. For using the ReproQ in quality improvement, the questionnaire should be able to identify best and worst practices. To achieve this, ReproQ should be reliable and able to identify relevant differences.

**Methods and Findings.** We sent questionnaires to 17,867 women six weeks after labor (response 32%). Additionally, we invited 915 women for the retest (response 29%). Next we determined the test–retest reliability, the Minimally Important Difference (MID) and six known group comparisons, using two scorings methods: the percentage women with at least one negative experience and the mean score. The reliability for the percentage negative experience and mean score was both 'good' (Absolute agreement = 79%; intraclass correlation coefficient = 0.78). The MID was 11% for the percentage negative and 0.15 for the mean score. Application of the MIDs revealed relevant differences in women's experience with regard to professional continuity, setting continuity and having travel time.

**Conclusions.** The measurement characteristics of the ReproQ support its use in quality improvement cycle. Test–retest reliability was good, and the observed minimal important difference allows for discrimination of good and poor performers, also at the level of specific features of performance.

# INTRODUCTION

Client experiences are considered to be important independent indicators for health care performance (*Valentine, Bonsel & Murray*, *2007*; *Valentine et al.*, *2003*). Being relevant for its own sake, client experiences also affect clinical outcome through several pathways

Corresponding author
Henk F. van Stel,
H.vanStel@umcutrecht.nl

(*Campbell, Roland & Buetow*, *2000*; *Sitzia & Wood*, *1997*; *Wensing et al.*, *1998*; *Williams*, *1994*). For example, clients who truly understand the explanation of their caregiver are more likely to comply to treatment or to change lifestyle, and arguably a patient-unfriendly clinical staff or an intimidating hospital-setting will not support recovery.

The routine measurement and use of client experiences play an indispensable role in systematic quality improvement (*Haugum et al.*, *2014*; *Weinick et al.*, *2014*). For that purpose, the client information can be used in a two-stage quality cycle. In the first stage, care providers that perform above or below average are identified. This process is also called benchmarking (*Department of Health*, *2010*; *Ellis*, *2006*; *Ettorchi-Tardy, Levif & Michel*, *2012*; *Kay*, *2007*). In the second stage, assumed underperformers are invited to improve their results followed by an internal quality cycle, where above-average performers ('best practices') may give guidance. Translated technically, the quality cycle starts with quantification of individual client experiences and clinical outcomes (casemix-adjusted), followed by the ranking across providers. Next, after defining thresholds, under- and best-performing units are defined. Finally, client experiences and other outcomes are analyzed in more detail. Preferably this break down of data is combined with face-to-face interactions among professionals. This more refined analysis offers tangible targets for improvement, unlike the global outcomes used in benchmarking.

To include clients' experiences with maternity care in the routine care quality evaluation and quality improvement, and in view of professionals, clinical organizations and health insurance companies who urged for measuring quality from the perspective of the client, we developed the Repro Questionnaire (ReproQ) (*Scheerhagen et al.*, *2015b*). This integral measure covers the period from first antenatal intake up to the postpartum period. The ReproQ consists of eight domains (33 experiences items), following the so-called WHO Responsiveness model (*Valentine, Bonsel & Murray*, *2007*; *Valentine et al.*, *2003*). All items strictly focus on service delivery from the clients' perspective.

Previously we demonstrated the feasibility, internal consistency and construct validity of the ReproQ (*Scheerhagen et al.*, *2015b*). In current paper we focus on psychometric properties needed to assess the suitability of the ReproQ for the two-stage quality improvement process. This suitability rests on two pillars: (1) are the judgments of pregnant women reliable, or stated otherwise, if the client survey is repeated do we get the same average and the same ranking of units; and (2) if we observe a quantitative difference between the average judgments of two care units—say 0.2 ReproQ points in our case—is this difference a relevant one? Epidemiologists developed a robust method to decide which differences are relevant in case of difficult to grasp clinical outcomes, the so-called minimal important difference (MID) approach. We tested these properties of the ReproQ to establish whether the ReproQ is suitable for both global benchmarking with a summary score (hence the assignment of below- and above-average performance), and for detailed profiling of providers or units or client groups once the underperforming units of client groups have been identified using the MID.

The data presented were collected during the provisional implementation of ReproQ measurement in about 1/3 of all perinatal units (hospitals with nearby midwife practices) in the Netherlands between October 2013 and January 2015. In 2015, the ReproQ was

appointed as a national client experience questionnaire and will be added to the national list of indicators in maternity care (*CPZ*, *2015*). Before the ReproQ was added to this national list, indicators only measured clinical outcomes (e.g., mortality, morbidity or complications) or parameters of professional performance. Adding the ReproQ to this list of indicators meets the WHO's recommendation to measure performance of health care systems also from the client's perspective. As indicator of performance the ReproQ should meet the conditions for a successful quality improvement cycle. This study explores two of these conditions: the ReproQ's reliability of the performance measurements and the MID as an aid to identify relevant differences between clients or perinatal units. The focus in this paper is on client's experiences with labor, because this is the key-event in maternity care. Antenatal care aims to create the best possible situation or starting point for labor. Antenatal risk assessment will be performed, and if necessary preventive measures and treatment of these risks are embedded. Provision of postnatal care is provided to reflect the outcome of the delivery for mother and child. Additionally, care during delivery is comparable in most Western countries, while antenatal and postnatal care are subject to more variation across countries or health systems.

## MATERIALS & METHODS

### Repro questionnaire

The questionnaire consists of two analogous versions: version A covers the experiences during pregnancy (antenatal) and version B covers the experiences during birth and the postnatal period. Version A is presented at about eight months gestational age, version B about six weeks postpartum. Each version asks for experiences at two instances, in case of version B postpartum experiences during labor, and experiences in the subsequent postpartum week respectively. As questions only differ with respect to the context referred to (say, experienced respect is asked for first antenatal visits, late in pregnancy, during labor, and during post-partum care), the resulting dataset represents a similar measurement covering four time intervals. In this article we focus on data from version B on the experiences during labor, the 3rd time point.

The 8-domain WHO responsiveness model is the conceptual basis of the ReproQ. Responsiveness is the way a client is treated by the professional and the environment in which the client is treated. Responsiveness is operationalized as four domains represent interactions with health professionals (dignity, autonomy, confidentiality, and communication), and four domains that reflect experiences with the organizational setting (prompt attention, access to family and community support, quality of basic amenities, and choice and continuity of care) (see Table 1) (*Valentine, Bonsel & Murray*, *2007*; *Valentine et al.*, *2003*). The response mode of the experience items uniformly used four categories: "never," "sometimes," "often," and "always," with a numerical range of 1 (worst) to 4 (best). An additional question which two domains are considered the most important, allows for a personalized scoring. Additional questions provide information on: (1) the rating of the global experience; (2) the process of care process, the location of care (e.g., home or hospital) and the primary health professional being responsible (e.g., midwife or obstetrician); (3)

**Table 1** Description of the eight WHO Responsiveness domains.

| Domain | Description |
| --- | --- |
| Dignity | Receiving care in a respectful, caring, non-discriminatory setting. |
| Autonomy | The need to involve the individuals in the decision-making process to the extent that they wish this to occur; the right of patients of sound mind to refuse treatment for themselves. |
| Confidentiality | The privacy of the environment in which consultations are conducted by health providers; the confidentiality of medical records and information about individuals. |
| Communication | The notion that providers explain clearly to the patient and family. The nature of the illness, and details for the required treatment and options. It also includes providing time for patients to understand their symptoms and to ask questions |
| Prompt Attention | Care provided readily or as soon as necessary |
| Social considerations | The feeling of being cared for and loved, valued, esteemed and able to count on others should the need arise. |
| Basic Amenities | The extent to which the physical infrastructure of a health facility is welcoming and pleasant |
| Choice and Continuity | The power or opportunity to select, which requires more than one option. |

the clinical outcome of both mother and child, as perceived by the mother; (4) information about previous pregnancies; and (5) client's socio-demographic characteristics.

Content validity of the ReproQ-version-0 was determined through structured interviews with pregnant women, women who recently had given birth, and health care professionals. All Responsiveness domains were judged relevant. Construct validity of the adapted ReproQ-version-1 was determined through a web-based survey, and based on response patterns; exploratory factor analysis; association of the overall score with a Visual Analogue Scale; and known group comparisons. The exploratory factor analysis supported the assumed domain structure and suggested several adaptations. Correlation of the VAS rating and overall ReproQ score supported validity for the antenatal and postnatal versions of the ReproQ. Further details are described elsewhere (*Scheerhagen et al.*, *2015b*).

## Data collection

In current study, data were obtained from three sources. The majority of data were collected by three postnatal care organizations (organizations that deliver postnatal care over a period of seven to 10 days). Additional data were collected by the national Birth Centre Study (a university-based research organization), and from 10 perinatal units (a hospital with associated midwifery practices). There were no exclusion criteria regarding organization, health care professional or client.

Data collection implied that clients were invited to participate by their care provider on behalf of the research team. With their consent, name and e-mail address were obtained and provided to the organization that distributed the digital survey. Women provided formal informed consent at the beginning of the questionnaire. For the Birth Centre Study and 10 perinatal units, the research team received client's name and e-mail information for recruitment after written informed consent had been obtained. The person who included the woman can, theoretically, be the same as the health care professional in charge of the

delivery (usually gynecologist or community midwife), but this is highly unlikely to be the case and not typical of our obstetric care system.

During data collection, an extensive data privacy protocol applied. The Medical Ethical Review Board of the Erasmus Medical Center, Rotterdam, the Netherlands, approved the study protocol (study number MEC-2013-455).

Data were collected in two waves. The first wave was between October 2013 and January 2015. Six weeks after the expected date of labor, all participating women received an invitation to fill out the postnatal ReproQ questionnaire. Non-responding women received a reminder two weeks later. These data were used to determine the MID and compare the known groups. The second wave occurred during October 2014 and January 2015. All women who previously filled out the postnatal ReproQ measurement in the first wave were invited to fill out their experiences again for the test–retest comparison. Excluded from invitation were women whose answers in the postnatal ReproQ were largely incomplete. The intended test–retest interval was 14 days. Since women's situation might change during the test–retest interval, we added the following item for verification. "*Have you experienced something important in the last two weeks?*"

## Participating women

Sample size was not formally calculated since we had no prior data to use as input data. Additionally, a formal sample size calculation seems questionable since statistical testing does not a play role in the estimation of the MID. Moreover, we anticipated that the provisional national implementation of this survey would provide sufficient numbers of responses for the study questions. For the MID and known groups comparison, we included all usable responses. For the test–retest, we aimed at 200 usable questionnaires.

In the first wave, we invited 17,867 women who recently had given birth, of whom 5,760 responded to the survey (32%). We excluded 877 women, because they filled out less than two of the following characteristics: ethnicity, educational level, care process, and experienced outcome of the mother and baby. We considered these background data as critical to describe the study participants in sufficient detail, and to understand and interpret the ReproQ scores and the associated MIDs. In the second wave, we invited 915 women for the retest, of whom 265 responded (29%). We excluded 57 women for the retest, because their situation changed negatively or was unknown. We did so because, a test–retest analysis requires that context and conditions between the test and retest situations remain unaltered (*De Vet et al.*, *2011*). To judge representativeness, we compared the characteristics of 208 women in the test–retest with the 4,675 women who filled out the test once using standard Chi square tests.

## ReproQ score model

We used two scoring models to summarize women's experiences: the proportion women with negative experience(s) (in short: 'percentage negative') and the mean score. Both were calculated for the eight individual domains, the four personal domains, the four setting domains and a total score across all domains. Percentage negative was defined as filling out the response category 'never' in at least one of the domains and/or filling out 'sometimes'

in a domain that the client identified as most important. The percentage negative method avoids compensation of a negative experience by positive experiences on other items of domains, whereas the mean scores allow the compensation of negative experiences. The mean scores were computed as unweighted average-scores, treating never (1), sometimes (2), most of the time (3) and always (4) numerically.

## Data analysis
### Test–retest reliability

Test–retest reliability was assessed using three measures. (1) For the percentage negative, we used the percentage absolute agreement, classified as excellent' (90%–100%), 'good' (75%–89%), 'moderate' (60%–74%), or 'poor' (<60%) (*Singh et al.*, *2011*). (2) For the mean scores, we used the Intraclass Correlation Coefficient (two way mixed model, absolute agreement, single average), classified as: 'excellent' ($\geq$.81), 'good' (.61–.80), 'moderate' (.41–.60), 'poor' ($\leq$.40) (*Singh et al.*, *2011*). (3) Finally we created the Bland–Altman plot, calculating the bias (or mean difference between test and retest scores) and the limits of agreement, equal to the mean difference $\pm$ 2*SD of that mean difference (*Bland & Altman*, *1986*; *Streiner & Norman*, *2008*).

### Minimally important difference

We determined the MID using (1) the anchor-based (or the difference in score between two adjacent levels of an anchor-question (*Copay et al.*, *2007*)) and (2) distribution-based method (or the difference in distribution of observed scores (*Revicki et al.*, *2008*)), each having their merits.

As anchor-question we used the global rating of a client's experience: "*Overall, how would you rate the care received during your labor and care after birth?*" (in short: 'Global rating'). This anchor-question emerged as best option in a review study of the Picker Institute (*Graham & Maccormick*, *2012*). Women could respond to this question on a 10-point VAS. We determined the mean score and the percentage negative of the individual domains, personal, setting and total scores for the VAS ratings 7, 8 and 9. We used the global rating of '8' as reference category, being the mode in our data (*Copay et al.*, *2007*). Next, the MID was calculated by subtracting these mean scores of the adjacent categories 7 and 9 from the mean score of the reference, being 8, to check if the differences 7–8 and 8–9 were equal (*Copay et al.*, *2007*). The same procedure was used to calculate the MID of the percentage negative. The distribution-based MID was only calculated for the mean score. To determine the MID with distributed-based methods, we calculated the standard error of measurement (SEM) (*Wyrwich, Tierney & Wolinsky*, *2002*), and one half of the standard deviation ($\frac{1}{2}$SD) (*King*, *2011*; *Norman, Sloan & Wyrwich*, *2003*). The SEM is estimated by the baseline SD of the measurement multiplied by the square root of 1 minus its reliability coefficient (ICC from the test–retest assessment) (*Rejas et al.*, *2011*; *Vernon et al.*, *2010*; *Wyrwich, Tierney & Wolinsky*, *2002*). A difference larger than 1 SEM is thought to indicate a true difference between groups (*Copay et al.*, *2007*; *Revicki et al.*, *2008*). The $\frac{1}{2}$SD margin is regarded as a relevant difference as well (*Copay et al.*, *2007*; *Norman, Sloan & Wyrwich*, *2003*; *Revicki et al.*, *2008*).

*Clinical known-group comparison*

We used six so-called known group comparisons (in terms of clinical outcome) to assess the discriminative validity of the ReproQ. Here we determine if women from different 'known groups' also have different mean experience scores and percentage negative (setting, personal, overall), and if these differences exceed the anchor-based MIDs for 7–8 and 8–9.

We made the following 'known groups': First, we compared the scores of women who before the labor did and did not meet the health care professional who supervised their labor, this being a proxy of professional continuity (*Saultz & Lochner*, *2005*). Second, for setting continuity, we compared the scores of women who were entirely low risk versus women who shifted from low-risk to high-risk during parturition. These women have the highest mortality and morbidity risk (*Evers et al.*, *2010*; *Poeran et al.*, *2015*). Third, we compared the scores of women who started their labor in office hours (8:00 am–5:00 pm, Mondays to Fridays) versus past office hours (*Gould, Qin & Chavez*, *2005*; *Gould et al.*, *2003*; *Stephansson et al.*, *2003*; *Urato et al.*, *2006*). Fourth, we compared the scores of women who had to travel 15 min or more with women who had to travel less than 15 min. In agreement with literature, we only included women in this comparison who were transferred from home to hospital during parturition and whose birth was unplanned (*Poeran et al.*, *2014*; *Ravelli et al.*, *2011*). Fifth, we compared the scores of women who had an emergency with women who had a planned caesarean section (*Elvedi-Gasparovic, Klepac-Pulanic & Peter*, *2006*). Finally, as proxy of concentration of care, we compared the scores of women who delivered in small hospitals (<750 labors annually (first quartile)) vs. large hospitals ($\geq$1500 labors annually (fourth quartile)) (*Finnstrom et al.*, *2006*; *Moster, Lie & Markestad*, *1999*; *Moster, Lie & Markestad*, *2001*; *Phibbs et al.*, *1996*; *Tracy et al.*, *2006*).

## RESULTS

Table 2 presents the characteristics of responding women who filled out the test ($n = 4,675$) and women who filled out the retest ($n = 208$). Mean age was 31 years (SD = 4.3); 398 (8%) women were of non-Western background; and 368 (8%) women reported to have a low educational level (both percentages slightly below national average). About half of the women gave birth for the first time (52%; about national average), and 2,313 (48%) women did not know the health care professional who supervised labor. 527 (11%) women were referred to secondary care during their pregnancy; 1,724 (36%) were referred during parturition (about the national average) and 618 (12%) women had a cesarean section (below the national average of 18%). The characteristics of women who filled out the retest differed significantly in terms of ethnic background (more Western women), setting continuity (more women were referred to secondary care during pregnancy), and global rating (women gave a higher global rating).

### Test–retest reliability

Table 3 shows the test–retest reliability. All experience items combined, 47% of the women reported one or more negative experiences filling out the test. When filling out the retest, 40% of women reported one or more negative experiences. The absolute test–retest agreement of 'having a negative experience' was 78.8% CI [72.6–84.2]. The ICC of the total scores

**Table 2** Characteristics of women who filled out the test ($n = 4,675$) and the retest ($n = 208$).[a]

| | | Test (%) | Retest (%) |
|---|---|---|---|
| **Socio-demographics** | | | |
| Age | ≤24 years | 6 | 4 |
| | 25–29 years | 30 | 31 |
| | 30–34 years | 42 | 48 |
| | ≥35 years | 22 | 17 |
| Parity | Primiparous | 52 | 48 |
| Ethnic background[b] | Non-Western | 9 | 3 |
| Educational level | Low | 8 | 5 |
| | Middle | 35 | 33 |
| | High | 57 | 61 |
| Marital status | Married/living together | 96 | 97 |
| | Relationship, not living together | 2 | 2 |
| | No relationship | 2 | 2 |
| **Care** | | | |
| Professional continuity | No | 48 | 51 |
| Setting continuity | Primary care only | 37 | 34 |
| | Secondary care only | 16 | 15 |
| | Referral to secondary care during pregnancy | 11 | 17 |
| | Referral to secondary care during parturition | 36 | 34 |
| Onset of delivery | Outside office hours | 70 | 64 |
| Travel time[b] | None or by choice | 70 | 79 |
| | <15 min during delivery | 18 | 14 |
| | ≥15 min during delivery | 12 | 7 |
| Cesarean section | No | 87 | 87 |
| | Planned cesarean | 4 | 3 |
| | Emergency cesarean | 8 | 10 |
| Hospital size of the perinatal unit | <750 deliveries per year | 12 | 11 |
| | 750–1,499 deliveries per year | 47 | 44 |
| | ≥1500 deliveries per year | 40 | 46 |
| **Quality** | | | |
| Picker overall rating[b] | ≤6 | 8 | 4 |
| | 7 | 16 | 8 |
| | 8 | 34 | 39 |
| | 9 | 26 | 30 |
| | 10 | 16 | 18 |

**Notes.**
[a] The percentage of missing data was below 5% in all characteristics, and will therefore not be presented.
[b] Significant difference between the participating women of the test and women participating the retest.

(mean$_{test}$ = 3.79; mean$_{retest}$ = 3.78) was 0.78, showing good reliability. The mean test–retest difference of the total score was 0.01; limits of agreement were +0.31 and −0.31. The reliability of the personal and setting scores was similar to the reliability of the total score.

The level of agreement regarding negative experiences within individual domains was excellent, except for the domains Autonomy and for Choice and Continuity that showed

**Table 3** Test–retest reliability of the experience during labor, on percentage women with a negative experience and mean score (*n* = 208).

| Score | Negative experience[a] | | | Mean experience | | | | |
|---|---|---|---|---|---|---|---|---|
| | Test (%) | Retest (%) | Absolute agreement (%) | Test mean (SD) | Retest mean (SD) | ICC | Bias | Limits of agreement[b] |
| **Overall** | **46.6%** | **39.9%** | **78.8%** | **3.79 (0.21)** | **3.78 (0.23)** | **0.78** | **0.01** | **0.31** |
| **Personal domains** | **31.7%** | **27.4%** | **82.2%** | **3.75 (0.27)** | **3.74 (0.28)** | **0.74** | **0.02** | **0.42** |
| **Setting domains** | **25.0%** | **22.6%** | **83.2%** | **3.82 (0.21)** | **3.83 (0.22)** | **0.74** | **−0.01** | **0.31** |
| Dignity | 3.4% | 2.9% | 94.7% | 3.89 (0.23) | 3.84 (0.27) | 0.62 | 0.05 | 0.48 |
| Autonomy | 27.9% | 26.0% | 86.5% | 3.46 (0.59) | 3.50 (0.50) | 0.65 | −0.04 | 0.88 |
| Confidentiality | 1.9% | 1.4% | 96.6% | 3.84 (0.36) | 3.81 (0.35) | 0.49 | 0.02 | 0.74 |
| Communication | 2.4% | 1.4% | 98.1% | 3.81 (0.34) | 3.79 (0.34) | 0.70 | 0.02 | 0.55 |
| Prompt Attention | 6.7% | 4.8% | 94.2% | 3.81 (0.30) | 3.82 (0.29) | 0.64 | −0.01 | 0.49 |
| Social considerations | 2.4% | 1.9% | 98.6% | 3.89 (0.28) | 3.90 (0.24) | 0.54 | −0.01 | 0.49 |
| Basic Amenities | 1.4% | 1.0% | 99.5% | 3.89 (0.25) | 3.89 (0.27) | 0.58 | 0.05 | 0.48 |
| Choice and Continuity | 18.3% | 18.3% | 85.6% | 3.69 (0.44) | 3.69 (0.45) | 0.62 | 0.00 | 0.78 |

**Notes.**
[a] Most negative experience (never) in a domain and/or 'sometimes' in the individually chosen two most important domains.
[b] The Bland Altman plot of the total score are presented in File S2.

good agreement. In these two domains, women also reported a higher level of negative experiences (Autonomy: 27.9%; Choice and Continuity: 18.3%) than in other domains (<7%). The ICCs varied between moderate (0.49 for Confidentiality) and good (0.70 for Communication). The bias was minimal (≤0.05) and was highest in the domains Dignity and Basic Amenities.

## Minimally important difference

Table 4 shows the MID results, using the two scoring models, including the results for the 7–8 and 8–9 differences. Using the percentage negative experience, the MID was 11.0%, based on the difference between the global ratings of 7 and 8. This means that the respondents rating their overall experience with the global rating scale with 7 showed 11% more cases of negative experiences compared to the respondents with the rating 8. When comparing the rating of 8 with 9, the MID was 9.2%. If we focus on the personal score, the MID using the 7–8 difference was 8.5%, which was comparable to the MID of 8.9% using the 8–9 difference. For the setting score, the MID 7–8 was 5.4%, which was smaller than the MID 8–9 (6.2%). The MIDs of the individual domains were all ≤8%. Using the ReproQ overall mean instead of the percentage negative experiences, the anchor-based MID based on the 7–8 distance was 0.15; when based on the 8–9 rating difference the MID was 0.10. The mean-MIDs of the personal score were slightly larger than the mean-MIDs of the setting score, and the domain MIDs showed some heterogeneity; both patterns were also observed using MIDs for negative experiences.

The use of the mean score also allowed the computation of a distribution-based MID. The distribution-based mean-MIDs of the 7–8 differences of the personal, setting and total score were similar to the anchor-based MIDs. In case of the individual domains, all distribution-based mean-MIDs were a somewhat larger than the anchor-based mean-MIDs.

**Table 4 Minimally important difference of the experience during labor based on the mean scores and the percentage women that had a negative experience ($n = 3,841$).[a]**

| Global rating per score | N | Negative experience[b] | | Mean experience | | | | |
|---|---|---|---|---|---|---|---|---|
| | | % Neg | Anchor based MID | Mean | Anchor based MID | SD | Distribution-based MID | |
| | | | | | | | SEM | ½SD |
| **Total score** | | | | | | | | |
| 7 (→8) | 584 | 60.4% | 11.0% | 3.59 | 0.15 | | | |
| 8 (ref) | 1,322 | 49.4% | | 3.74 | | 0.29 | 0.14 | 0.14 |
| 9 (←8) | 1,021 | 40.2% | 9.2% | 3.84 | 0.10 | | | |
| **Personal score** | | | | | | | | |
| 7 (→8) | 584 | 44.9% | 8.5% | 3.52 | 0.17 | | | |
| 8 (ref) | 1,322 | 36.4% | | 3.69 | | 0.35 | 0.18 | 0.17 |
| 9 (←8) | 1,021 | 27.5% | 8.9% | 3.80 | 0.11 | | | |
| **Setting score** | | | | | | | | |
| 7 (→8) | 584 | 36.6% | 11.6% | 3.66 | 0.13 | | | |
| 8 (ref) | 1,322 | 25.0% | | 3.79 | | 0.28 | 0.14 | 0.14 |
| 9 (←8) | 1,021 | 18.8% | 6.2% | 3.87 | 0.08 | | | |
| **Dignity** | | | | | | | | |
| 7 (→8) | 584 | 11.6% | 6.9% | 3.66 | 0.18 | | | |
| 8 (ref) | 1,322 | 4.7% | | 3.84 | | 0.34 | 0.21 | 0.17 |
| 9 (←8) | 1,021 | 2.0% | 2.7% | 3.93 | 0.09 | | | |
| **Autonomy** | | | | | | | | |
| 7 (→8) | 584 | 36.0% | 5.4% | 3.22 | 0.17 | | | |
| 8 (ref) | 1,322 | 30.6% | | 3.39 | | 0.58 | 0.35 | 0.29 |
| 9 (←8) | 1,021 | 25.6% | 5.0% | 3.56 | 0.17 | | | |
| **Confidentiality** | | | | | | | | |
| 7 (→8) | 584 | 5.7% | 2.0% | 3.64 | 0.17 | | | |
| 8 (ref) | 1,322 | 3.7% | | 3.80 | | 0.46 | 0.33 | 0.23 |
| 9 (←8) | 1,021 | 1.8% | 1.9% | 3.88 | 0.08 | | | |
| **Communication** | | | | | | | | |
| 7 (→8) | 584 | 5.3% | 2.4% | 3.55 | 0.17 | | | |
| 8 (ref) | 1,322 | 2.9% | | 3.73 | | 0.40 | 0.22 | 0.20 |
| 9 (←8) | 1,021 | 0.7% | 2.2% | 3.84 | 0.11 | | | |
| **Prompt Attention** | | | | | | | | |
| 7 (→8) | 584 | 10.3% | 4.4% | 3.62 | 0.13 | | | |
| 8 (ref) | 1,322 | 5.9% | | 3.75 | | 0.35 | 0.35 | 0.18 |
| 9 (←8) | 1,021 | 3.9% | 2.0% | 3.85 | 0.10 | | | |
| **Social Considerations** | | | | | | | | |
| 7 (→8) | 584 | 3.3% | 1.0% | 3.75 | 0.12 | | | |
| 8 (ref) | 1,322 | 2.3% | | 3.87 | | 0.33 | 0.22 | 0.16 |
| 9 (←8) | 1,021 | 1.5% | 0.8% | 3.91 | 0.04 | | | |
**Table 4** (*continued*)

| Global rating per score | N | Negative experience[b] | | Mean experience | | | | |
|---|---|---|---|---|---|---|---|---|
| | | % Neg | Anchor based MID | Mean | Anchor based MID | SD | Distribution-based MID | |
| | | | | | | | SEM | $\frac{1}{2}$SD |
| **Basic Amenities** | | | | | | | | |
| 7 (→8) | 584 | 4.6% | 2.5% | 3.80 | 0.07 | | | |
| 8 (ref) | 1,322 | 2.1% | | 3.87 | | 0.30 | 0.20 | 0.15 |
| 9 (←8) | 1,021 | 1.6% | 0.5% | 3.92 | 0.05 | | | |
| **Choice and Continuity** | | | | | | | | |
| 7 (→8) | 584 | 26.2% | 8.0% | 3.46 | 0.19 | | | |
| 8 (ref) | 1,322 | 18.2% | | 3.65 | | 0.52 | 0.32 | 0.26 |
| 9 (←8) | 1,021 | 13.3% | 4.9% | 3.78 | 0.13 | | | |

**Notes.**
[a]Due to a software problem this item was not presented to 20% of the participating women.
[b]Most negative experience (never) in an domain and/or 'sometimes' in the individually chosen 2 most important domains.

## Clinical known-group comparison

Figure 1A shows the impact of six known groups with an assumed influence on client experiences, using the percentage of negative experiences as scoring model. Two out of six comparisons showed differences in agreement with expectations. Already knowing the professional who supervised labor (i.e., continuity of professional), had a considerable impact: the differences in total score and personal score of women who knew and did not know their professional were larger than the associated MIDs (7–8 difference). Similarly, referral during labor (i.e., discontinuity of setting) was associated with differences in total, personal and setting scores larger than the MID.

Figure 1B shows the same known groups comparison, now using the mean ReproQ scores and the associated MID. The difference in mean overall, setting and personal scores between women who received only primary care and women who were transferred during parturition was larger than the corresponding MIDs (7–8 difference). All three differences scores of personal continuity and setting continuity were larger than the MIDs (8–9 difference). Further details are presented in File S1.

## DISCUSSION

To determine the suitability of ReproQ in the two-stage quality improvement cycle, we assessed its test–retest reliability and determined the MID according to two methods. Test–retest reliability was good for both scoring models. The anchor-based MID of the percentage negative experiences was 11%; the anchor-based MID of the mean score was 0.15 (on a range of 1–4). The distribution-based MIDs (SEM) proved about similar to the anchor-based mean-MID of the overall, personal and setting scores. However, for the domain scores the SEM exceeded the anchor-based mean-MIDs. The known-group comparisons showed that knowing the professional that supervised your labor and not being referred during labor had considerable impact on the experiences scores. As the

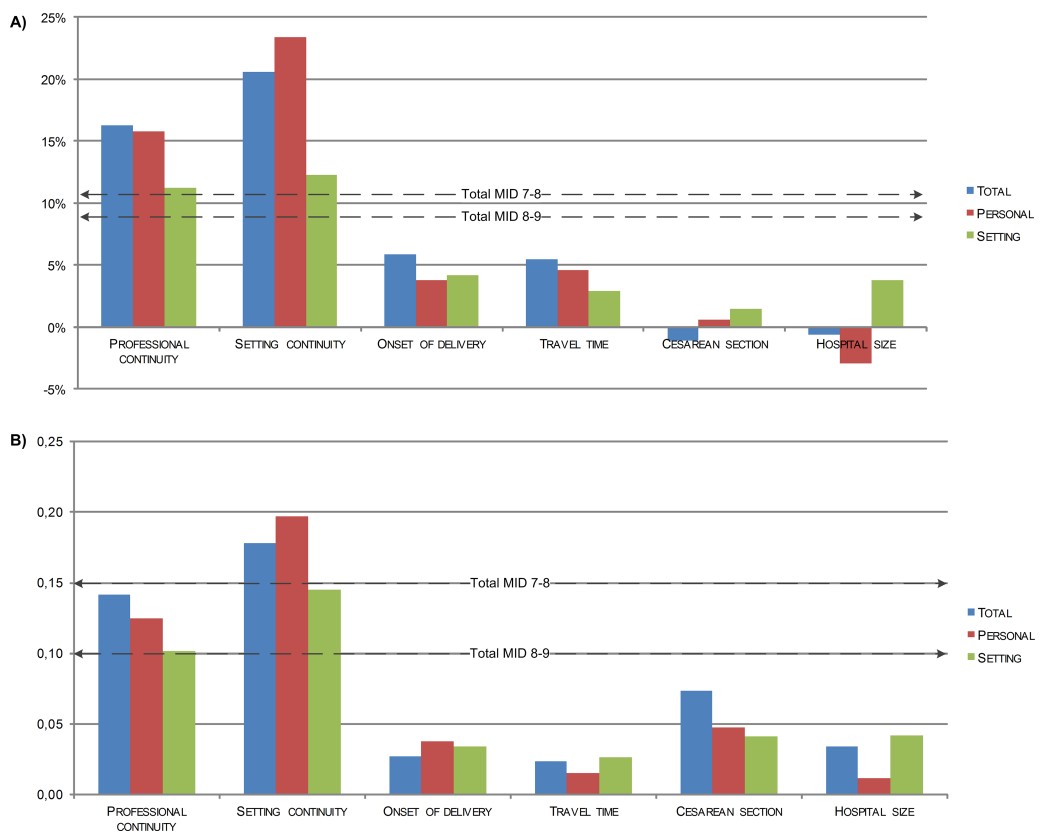

**Figure 1** **Difference in ReproQ in terms of negative score (A) and mean score (B), between the least preferred and the most preferred state, in 6 known-groups ($n = 4,883$).** Professional continuity—difference supervisor of the delivery is known vs. unknown (52%/48%) Setting continuity—difference primary care only vs. referred during labor (37%/36%) Onset of delivery—difference in vs. outside office hours (30%/70%) Travel time—difference women had to travel <15 min vs. ≥15 min, when transferred from home to hospital during labor (17%/11%). Cesarean section—difference planned vs. emergency cesarean section (4%/8%) Hospital size—difference < 750 deliveries per year vs. ≥ 1,500 deliveries per year (3%/12%).

observed ReproQ scores deviated more than the MID, we believe this instrument can be used as a benchmark with an interpretation of meaningful differences beyond statistical significance. Thus, the ReproQ can successfully identify areas that need improvement in subgroups of clients. One should be aware that the MID cannot be used to identify changes in (poor) experiences within clients.

Applying the percentage negative on the test–retest reliability showed that the reliability of the domains was higher than for the summery scores. This is surprising, because the likelihood to report a negative experience in both the test and retest is considerably larger for the summary scores than for the domains.

For the individual domains, fewer women reported a negative experience when filling out the retest than the test. The domains Autonomy and Choice and Continuity showed similar percentages of negative experiences in the test and retest, though the reliability of these domains was low compared to the other domains. This indicates that women who reported a negative experience filling out the test are not the same women that reported

a negative experience filling out the retest. Possible explanations for these effects are recall bias and/or response shifts, e.g., women adjust their opinion due to sharing their experiences with family and friends.

The summary scores showed higher reliability than the domain scores when using the mean score method. The explanation is that ICCs invariably increase when the summary scores include more items. When calculating a summary score, differences within a domain can be compensated by differences between the domains. This increases the stability of the summery scores.

The reliability of the domains Confidentiality and Social considerations was somewhat lower than for the other domains. It is possible that women feel that these concepts are difficult to judge, which increases the fluctuations in domain scores.

Both the negative-MID and the mean-MID varied across adjacent response categories. When the global rating increases neither the percentage negative decreases nor the mean scores increase linear in all scores. This suggests that a gain in client experience as the result of quality improvement is not similar to the loss in client experience as the result of deterioration. One explanation is that clients do not weigh all domains are equally in the global rating. Another explanation is that respondents are not inclined to use the extreme response categories.

The distribution-based MIDs (SEM) were similar to the anchor-based mean-MID of the overall, personal and setting scores. However, for the domain scores the SEM exceeded the anchor-based mean-MIDs, because the SDs of the domain scores were larger than the SDs of the summary scores, and because the domain ICC scores were lower.

The known-group comparisons were based on previously reported differences in clinical outcomes. Professional and setting continuity also resulted in large and relevant differences in experience scores. These differences are probably due to the deviation of the expected or planned process of care, which might result in a stressful event, even when this deviation is clinically necessary. Differences in experiences of the other clinical known-group comparisons were not relevant. It is possible that these experiences are not or only partly correlated with the clinical outcomes. Another explanation might be that the experiences were reported in retrospect. Perhaps women's experiences were biased afterwards by a good maternal or child outcome, by better, sufficient or intensive postnatal care when complications occurred during labor. It is also possible that women's experiences of the process of labor were affected by hormones and stress, or that women lacked information on what normal maternity care is. (Note that about half of the women was primiparous).

## Strengths

To our knowledge, this is the first study to clarify the meaning or relevance of score differences obtained with client experience questionnaires. So far, current studies towards the MID mainly focussed on quality of life scores (*Brozek, Guyatt & Schunemann*, *2006*; *Copay et al.*, *2007*; *Guyatt et al.*, *2002*). Secondly, the use of global ratings is debated due to their unknown validity and reliability (*Copay et al.*, *2007*). By using the overarching question of the British National Patient Survey Coordination Centre as anchor-question we met this critique: this global rating is extensively tested and has a good content and

construct validity (*Graham & Maccormick, 2012*). Thirdly, we explored the differences between 7–8 and 8–9 changes in global rating. By doing so, we were able to check the assumption that both differences were similar. Inevitably, preference scales are to some extent non-linear in interpretation, which applies to both the ReproQ and to scales used for anchoring. At the upper or lower ends of the scale the interpretation of gains and losses may differ, and the 'degree of relevance' of one step higher (7–8) or lower (8–9) decreases. Since benchmarking is usually based on the comparison of averages, the impact of non-linearity is probably small.

We previously introduced the percentage negative experiences as an alternative scoring to the frequently used mean score. Three remarks should be made. Firstly, we deliberately focused on the percentage negative experiences instead of percentage positive experiences. Focusing on the latter may contribute to the validity of the findings. However, from a practical perspective, we chose to emphasize the percentage negative experiences because in quality improvement cycles most benefit can be obtained when poor performing providers or centers are identified and improvements can be implemented. The percentage negative experiences therefore seems more relevant for quality improvement than the percentage positive experiences. We expect that the benefit of quality improvements for centers with a high percentage of positive experiences is less than the benefit for poor performing centers. Secondly, both the percentage negative experiences as well as the mean score can be used for benchmark purposes. Despite differences in approach, both may lead to the same identification of relevant differences in subgroups (see Fig. 1). Finally, one could argue that our approach of the MID is conservative, as it actually defines the size of a relevant minimal difference between averages on the group level on the base of differences in individual global ratings.

## Limitations

First, we sent the postnatal questionnaire six weeks after the expected date of labor, but it is unknown if this timing was optimal. An invitation later than six weeks could result in recall bias due to exposure to other influences (e.g., women return to work, assuming their usual habits and patterns), and/or in non-response because sharing one's birth experiences may seem less relevant. An invitation before six weeks is not necessarily a better option. It may result in better recollection of the experiences but the risk of mood swings and hormonal disturbances might affect responses and response rates.

Related to this: the postnatal questionnaire was not sent six weeks after the actual date of labor. We only had the expected date of delivery as anchor. To protect women's privacy, we were not allowed to collect the precise date of childbirth in the ReproQ. Since the expected date may deviate from the true date, women may have been surveyed earlier (but not more than two weeks earlier) when they delivered after the expected date, or about four to five weeks later for most women when they delivered before the expected date. In both cases, postnatal care had already ended, and it is unlikely that differences in timing of the invitation of the ReproQ may have resulted in different ReproQ scores between these groups.

Secondly, women with a low educational level and non-Western women were underrepresented despite considerable efforts to have them participate. Most likely this is selective non-response, as non-Western women report more negative experiences than Western women (*Scheerhagen et al.*, *2015a*), and non-Western women are more often low educated and/or more health illiterate (*Agyemang et al.*, *2006*; *Engelhard*, *2007*; *Fransen, Harris & Essink-Bot*, *2013*). Addition of the non-response group is likely to widen the gap between poor and good experiences. This does not necessarily affect the estimated MID. Our non-Western women reported both a lower ReproQ score as well as a lower global rating than Western women. Repeating the MID calculations without this subgroup (non-Western women with a low educational level) resulted in about similar results. When this subgroup was excluded, the percentage negative MID decreases maximum −0.1% and increases max. +1.2%. Similar, the mean MID varies −0.01 to +0.03. Hence, the underrepresentation of these subgroups has limited impact on the estimated MID. Regrettably, we could not find additional evidence on the influence of selection bias on psychometric properties of other client experience surveys with similar characteristics in terms of study population, length and mode of administration (e.g., read out loud by clinicians vs. stand alone, self-report) (*Rejas et al.*, *2011*; *Vernon et al.*, *2010*). The impact of care process, birth outcome and socio-demographics on experiences scores, test–retest reliability and MID requires further study.

Thirdly, the MID is often used to identify changes in a patient's situation over time (*Brozek, Guyatt & Schunemann*, *2006*; *Copay et al.*, *2007*; *Guyatt et al.*, *2002*). Given the small time window of the labor phase, it is unfeasible to validly assess changes in survey-based experiences within clients. Therefore, our MID estimates are based on cross-sectional comparisons. Our MID cannot be used to identify changes within a client, but only between health care providers, or within health care providers over time. These provider differences are more relevant than changes within clients for improving the quality of maternity care by the two-stage quality cycle.

Finally, we aimed at suitability of the ReproQ survey across countries, by using the universal WHO Responsiveness concept, by following an accepted strategy for survey development, and by avoiding any preferences towards providers, specific professionals or organizational structures. It is unclear if clients in other countries have the same importance ratings, experiences and MIDs as Dutch clients. Other self-report instruments in maternity care, such as the Women's Experience of Maternity Care Questionnaire of the NHS, overall indicate very good experiences (*Peterson et al.*, *2005*; *Redshaw & Heikkila*, *2010*; *Smith*, *2001*, *2011*). Therefore, the MID in other developed countries will probably be in about same magnitude as our MID estimates.

## Future use

The psychometrics of the ReproQ appear adequate for benchmarking for targeting quality improvement based on the profile of domain scores, and for monitoring of domain specific quality improvements. As part of a routine two-stage quality improvement cycle, as proposed by the ICHOM (*ICHOM*, *2015*), we can identify relevant differences between birth care units who perform better or worse. The MID based percentage negative

discriminates (known) groups better than the mean-MID. Furthermore, we recommend to use a multi-item questionnaire for benchmarking, such as the ReproQ, instead of a single-item benchmark: the reliability of a single-item benchmark is much lower and, unlike the ReproQ, single items are less effective in guiding specific improvements.

To increase the response rate, alternative modes of data collection should be explored. One suggestion is to invite women to directly fill out the questionnaire when waiting for their health care professional in the waiting room. Another suggestion to minimalize selection and response bias is to send all women the questionnaire including informed consent, without involvement of individual health care professionals. A third suggestion is to translate and provide the questionnaire in several languages for non-Western women.

Additionally, future use should pay attention to ethnicity and socio-economic background, beyond routine case-mix adjustment procedures. Adjustment always bears the risk that unintentionally worse experiences are neutralized, taking away the incentive for improvement.

With many benchmarking activities into place, the second part of the quality cycle urgently needs more attention and explicit implementation. Evidence-based routine quality cycles are still rare. Implementation requires true information-guided cycles in some detail. The benefit of such an approach has been demonstrated in the evaluation of innovations (*Haugum et al.*, *2014*; *Weinick et al.*, *2014*). The introduction of the MIDs in quality cycles may convince stakeholders that progress through innovation is meaningful.

## CONCLUSION

Maternity care is continuously developing, partly based on the measurement of client experiences. The ReproQ questionnaire, based on the WHO Responsiveness model, is suitable to be used in quality improvement cycles: we showed good test–retest reliability, and by determining the minimally important difference relevant differences can be identified.

## ACKNOWLEDGEMENTS

We are grateful for all maternity care organizations that took part in our study. We are especially grateful to Christa Cats, Hans Reinold and Ester van Dalen, who inspired the postnatal care organizations to participate in our research.

### Funding
This study was funded by Stichting Miletus (www.stichtingmiletus.nl). The funders had no role in study design, data collection and analysis, decision to publish, or preparation of the manuscript.

### Grant Disclosures
The following grant information was disclosed by the authors:
Stichting Miletus.

## Competing Interests

The authors declare there are no competing interests.

## Author Contributions

- Marisja Scheerhagen conceived and designed the experiments, performed the experiments, analyzed the data, contributed reagents/materials/analysis tools, wrote the paper, prepared figures and/or tables.
- Henk F. van Stel conceived and designed the experiments, performed the experiments, contributed reagents/materials/analysis tools, prepared figures and/or tables, reviewed drafts of the paper.
- Dominique J.C. Tholhuijsen conceived and designed the experiments, performed the experiments, analyzed the data.
- Erwin Birnie conceived and designed the experiments, analyzed the data, contributed reagents/materials/analysis tools, wrote the paper, prepared figures and/or tables, reviewed drafts of the paper.
- Arie Franx contributed reagents/materials/analysis tools, prepared figures and/or tables, reviewed drafts of the paper.
- Gouke J. Bonsel conceived and designed the experiments, contributed reagents/materials/analysis tools, wrote the paper, prepared figures and/or tables, reviewed drafts of the paper.

## Human Ethics

The following information was supplied relating to ethical approvals (i.e., approving body and any reference numbers):

The Medical Ethical Review Board of the Erasmus Medical Center, Rotterdam, the Netherlands, approved the study protocol (study number MEC-2013-455).

## Data Availability

The fully anonymized data is available at the Dutch Dataverse Network (a public repository of all Dutch universities): http://hdl.handle.net/10411/20363. Complete data files and study materials are digitally stored by the University Medical Centre Utrecht, for conditional access in case of questions concerning scientific integrity.

## Supplemental Information

Supplemental information for this article can be found online at http://dx.doi.org/10.7717/peerj.2092#supplemental-information.

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
