# Peer review of "Applicability of the ReproQ client experiences questionnaire for quality improvement in maternity care"

_PeerJ, doi:10.7717/peerj.2092_

## Round 0.1 · original submission · Major Revisions

The manuscript was well recieved by the reviewers. Nevertheless please see the comments of reviewer #1 that raises some major concerns regarding patients selection. Please address these comments and resubmit the manuscript for further consideration

·

Basic reporting

07/02/2016

To
Prof. Offer Erez
Academic Editor
PeerJ

Dear Prof. Erez
Enclosed please find a review of the manuscript entitled "APPLICABILITY OF THE REPROQ CLIENT EXPERIENCES QUESTIONNAIRE FOR QUALITY IMPROVEMENT IN MATERNITY CARE” which I am recommending for publication in PeerJ after a major revision. I have prepared a summary of the study and a list of issues that the authors may want to address.

Regards,

Salvatore Andrea Mastrolia M.D.
Department of Obstetrics and Gynecology
Azienda Ospedaliera Universitaria Policlinico di Bari
University of Bari “Aldo Moro”, Bari, Italy
Piazza Giulio Cesare 11, 70123
Bari, Italy
Tel +390805593583, Fax +390805592228
International Fellowship Program
Maternal Fetal Medicine Unit
Department of Obstetrics and Gynecology “B”
Soroka University Medical Center
Ben Gurion University of the Negev
Beer Sheva, Israel
E-mail mastroliasa@gmail.com


Peer J Manuscript ID: 2016:01:8590:0:0:REVIEW
Title: "APPLICABILITY OF THE REPROQ CLIENT EXPERIENCES QUESTIONNAIRE FOR QUALITY IMPROVEMENT IN MATERNITY CARE”

Summary: The authors investigated and discussed the value of Repro Questionnaire, a multi-item questionnaire created to assess healthcare quality in women in all the stages of pregnancies as well as during postpartum period.
The ReproQ consists of 8 domains (33 experiences items), and follows the so-called WHO Responsiveness model. All items strictly focus on service delivery from patients' perspective. The authors decided to focus the study on validity of this scoring system in determining an improvement in evidence-based routine quality cycles at the moment of delivery.
Their conclusion is that this questionnaire showed to be suitable in evaluating the quality improvement cycle due to its good test-retest reliability.

Experimental design

Please read General comments for the Author

Validity of the findings

Please read General comments for the Author

Additional comments

General comments: I read the manuscript with interest and here is a list of major concerns that the authors may want to address:

1) The authors are dealing with ReproQ that they cite as validated in previous publications as well as the WHO Responsiveness model. I did not find both of them in the article or the supplementary material. I would suggest them to give the reader a clear perception of the scores and models they are using adding them to the manuscript.

2) My main concern regards the methods and especially the patients selection. Please read my specific comments on the Methods section.

3) There are several grammar mistakes across the manuscript. I would suggest the authors to check it carefully. More than that, the linguistic level is suitable for publication.



Specific comments

Title: The title is appropriate.

Abstract: The abstract is well written.

Introduction: Well written and stating clearly the idea behind the study.
1) Line 53: "...client experiences play.." instead of plays
2) Line 67: organizations instead of organisations
3) Line 93-94: I think the author should not save words on this concept. The last paragraph of their introduction is what makes all the manuscript worthy to be written. Instead of inducing the reader to quickly remember himself what was anticipated by the authors regarding the use of the instrument that is the object of the study, I would remind it here. The purpose of the study has to be absolutely clear in my opinion and this is the exact place to state it. Here I would explain also in few words why the authors decided to relate their study to delivery instead of the rest of the pregnancy or the postpartum period.

Materials and methods:
1) Line 121-124: I am not sure this paragraph belongs to the methods section. This is maybe a result and, if one decides to use it, there should be a comparison between what literature reports and what was found in the present study. Otherwise is an element without discussion that does not need to be included, in my opinion.
2) Line 127-128: The authors refer to three maternity care organizations. They should be more specific on the type of organization. Are they different from the Birth Centre Study or the 10 hospitals that are listed in the following sentence of the manuscript? If yes, it should be stated clearly.
3) Line 131: The figure of the patients' care provider should be clarified. In the results section the authors analyze the results according to six comparison groups. One of them includes women who met or did not meet the person in charge of their delivery before. Is this care provider referring the person in charge of the delivery or is he someone else? Can the person referring the woman to the study be the same person that will be in charge of the delivery? If yes, this is extremely important, since it represents a sort continuity of care that might affect the patients' satisfaction.
4) Lines 140-147: Data were collected in two waves. The second wave started six weeks the expected date of delivery. This means that there can be a deviation from this date since not all women deliver in the expected date. This should be showed in the results.
5) Lines 146-147: Which is the extra item that was added for verification, in order to study the changes in the situation during the test-retest interval?
7) Lines 144-145: How were women included in the subset for retesting? What criteria were followed?
8) Lines 150-152: The authors state that "Sample size was not formally calculated, as we anticipated that the provisional national implementation of this survey would provide sufficient numbers of responses for the study questions". From where this derives? This is an extremely important statement since the reason for not performing power calculations should be very clear, otherwise the entire structure of the study risks to fall.
9) Lines 155-156: On which basis women were excluded if they were filling less than two characteristics? Was it arbitrary of is there any other explanation? In case please provide it.
10) Lines 158-159: The authors state that 57 women in the retesting group were excluded since their situation changed negatively or was unknown. I am afraid this introduced a bias in the study, otherwise there should be a specific reason why women were excluded, especially those whose situation changed negatively.
11) Lines 177-178: Please fix lining.
12) Lines 216-218: Same as concern #3 for this section.
13) Line 225: "a planned" instead of "an planned".
14) Line 226: Please define "travel during childbirth". I would anyway change childbirth to labor or delivery across the manuscript. It would be more appropriate.
15) Is there a specific reason why the authors did not compare women who were considered low risk to high risk for the entire pregnancy? Why did they only compare low risk patients to those changing their labeling to high risk during the index pregnancy?

Results:
1) Line 237: Know instead of known.
2) Maybe showing p values would make the results of easier understanding to the reader. I would add them also to Table 1, Figure 1 and 2.

Discussion: It is maybe too short, even shorter than introduction. I would suggest the author to discuss thoroughly the reasons for their findings.

Conclusions: This paragraph is well written.

Tables and supplementary material: The material provided by the authors is clear. I would add the p values to Table 1. Please see also my general comments regarding the presentation in extense of the questionnaire.

Figures: Quality is adequate. I would add p values to the comparisons.

·

Basic reporting

Clear English language.
Good introduction that is well relevant referenced .
Good structure.
Relevant, high quality pictures, well labelled and described.

Experimental design

It is an original primary research within the scope of the journal.
Reasearch question well defined, except it is unclear if the observed minimal important difference is targeting poor performers or centers.
This point needs to be clarified as well how to conclude.
.
The investigation performed to a high technical and ethical standard.

Unclear if according their methods can be replicated

Conclusion is well stated including strengths and limitations.

Validity of the findings

Impact and novelty were well assessed
There is focusing on the negative experience but no notion on the positive experience to analyze and reinforce the conclusions regarding the validity of the findings.
The 6 weeks post partum questionnaire can be still in the period affected by hormonal changes sleep deprivation as well mood swings- suboptimal
There is no reference to delivery course and outcome. More complicated cases with adverse outcome may affect the satisfaction reported, as opposed to uncomplicated cases. I would strongly recommend to perform such analysis in order to test the validity of the results. All these can be translated into low satisfaction rate reported among women who did not know their professional, referral to other center and longer travel time . This needs to be reported in the discussion and conclusions.
The low number of respondent and the primipara distribution population do not represent the entire population. Those who responded (only 32%-29%)) may be subjected to an extremely negative (and less positive) experience that may affect the motivation to respond such a report. This also may affect the MID score (negative experience is imprinted in the memory more than positive). Lastly, minority of non-western respondent -8% and low educational level – 8% - emphasizing that this population is not well represented as well the negative experience is underreported –as this population
Would recommend another way to collect such data- such as the 6 weeks post natal office visit for FU – when all women are waiting for their doctors in the waiting room. This should be mentioned in the limitations of the study.

Additional comments

It is an original primary research within the scope of the journal.
Reasearch question well defined, except it is unclear if the observed minimal important difference is targeting poor performers or centers.
This point needs to be clarified as well how to conclude.
.
The investigation performed to a high technical and ethical standard.

Unclear if according their methods can be replicated

Conclusion is well stated including strengths and limitations.

Overall important study that I recommend to publish with the above suggestions.

---

## Round 0.2 · accepted · Accept

The authors have addressed in a satisfactory manner the comments of the reviewers and this manuscript can be accepted for publication.
Prior to publication - please add in a supplementary file an elaborate description of the study design and methods to allow it replication by interested readers